# Development and Characterization of Novel Active Chitosan Films Containing Fennel and Peppermint Essential Oils

**Ting Liu [1,2], Jingfan Wang [2], Fumin Chi [1,*], Zhankun Tan [3] and Liu Liu [2,*]**

[1]  College of Food Science, Tibet Agriculture & Animal Husbandry University, Nyingchi 860000, Tibet, China; Ting_liu@126.com
[2]  College of Food Engineering and Nutrition Science, Shaanxi Normal University, Xi'an 710119, Shaanxi, China; wangjingfansnnu3@163.com
[3]  College of Animal Science, Tibet Agriculture & Animal Husbandry University, Nyingchi 860000, Tibet, China; tanzhankun@xza.edu.cn
*  Correspondence: chifumin@xza.edu.cn (F.C.); liuliu@snnu.edu.cn (L.L.); Tel.: +86-0894-5822924 (F.C.); Fax: +86-0894-5822924 (F.C.)

**Abstract:** The effects of fennel essential oil (FEO) and peppermint essential oil (PEO) on chitosan-based films were evaluated in this study. The results showed that the addition of FEO and PEO slightly increased the density and thickness, while significantly decreasing the moisture content, water swelling, and solubility properties. The color values ($L$, $a$, $b$, $\Delta E$ and whiteness index (WI)) of the composite films containing FEO and PEO changed obviously with a tendency toward yellowness, which was beneficial in resisting food decomposition caused by ultraviolet light. The differential scanning calorimetry (DSC) and fourier-transform-infrared (FTIR) results indicated that the addition of FEO and PEO affected the structure of the chitosan films, while the interaction between the chitosan and polyphenols in FEO and PEO established new hydrogen bonds and improved the thermal stability. The environmental scanning electron microscopy (ESEM) illustrated that the surfaces of the composite films containing FEO and PEO were smooth, but the cross-section was rougher than in pure chitosan film. Furthermore, the composite films containing FEO and PEO exhibited prominent antioxidant activity. In short, the novel active chitosan-based films with incorporated FEO and PEO present broad application prospects in fresh-cut meat or vegetable packaging.

**Keywords:** active packaging; composite film; fennel essential oil; peppermint essential oil; antioxidant activity

---

## 1. Introduction

In recent years, consumers have shown increasing interest in improving the quality and safety of fresh-cut meat and vegetables through packaging. Active packaging is a new type of packaging with preservation and barrier properties [1]. Adding some natural compounds or ingredients to packaging materials to prepare active packaging materials can significantly improve their antibacterial and antioxidation properties, while providing other functions that did not originally exist in the packaging system [2]. Due to environmental and consumer health concerns, current research regarding active packaging has focused on biodegradable materials and natural preservatives [3].

Chitosan is often used as active packaging material for food products due to its non-toxicity and biocompatibility [3]. Moreover, chitosan can inhibit bacteria, mold, yeast, and other microorganisms, while showing significant potential for utilization in food preservation [1]. However, the application of pure chitosan films is limited by its poor physical properties. Adding natural antioxidants and natural

antibacterial agents such as natural polyphenols and essential oils (EOs) to chitosan can enhance its physical and biological properties and expand its application in active food packaging [4,5].

EOs are extracted from natural plants and possess prominent antibacterial and antioxidant effects. The incorporation of natural EOs into pure chitosan films can significantly improve their antibacterial and antioxidant properties, as well as their water solubility and vapor-permeability [6]. Fennel (*Foeniculum vulgare* Miller) belongs to the family Apiaceae and is generally classified into two subspecies vulgare and piperitum. The common species is the vulgare [7]. Fennel essential oil (FEO) is commonly used as a flavoring agent, cosmetics, and pharmaceutical products [8]. Some published researchers reported that biodegradable gelatin–chitosan films incorporated with FEO applied to fish during chilled storage had an excellent effect on fish preservation [9]. The application of peppermint essential oil (PEO) is growing in popularity due to its exceptional sensory properties. PEO contains menthol, menthone, and other major bioactive substances, exhibiting various inhibitory effects on foodborne pathogenic bacteria and fungi with different principal components [10,11]. PEO can effectively prolong the shelf life of food products. Chaemsanit et al. [11] reported that activated carbon adsorbed PEO, extending the shelf life and improving the preservation of the post-harvest quality of dragon fruit.

Although several studies have investigated the chemical, physical, structural, and biological properties of chitosan films containing EOs, the reports regarding chitosan composite films containing fennel and peppermint remain limited. This study focuses on the physical, characterization and antioxidant properties of Chitosan-FEO, Chitosan-PEO, and chitosan composite films added with both EOs (Chitosan-F/P) to identify their potential to be used as active packaging for fresh-cut meat and vegetables.

## 2. Materials and Methods

### 2.1. Materials

Chitosan (degree of deacetylation $\geq$ 90%, molecular weight of around $1.5 \times 10^5$ Da, viscosity 100–200 mPa·s) was supplied by Aladdin Chemical Co. (Shanghai, China). FEO and PEO were purchased from LANMU Technology Co. (Beijing, China). All other reagents were of analytical grade and purchased from Jingbo Chemical Co. (Xi'an, China).

### 2.2. Preparation of Chitosan Films

The chitosan (2%, *w/v*) was added to acetic acid (1%, *v/v*) solution and stirred overnight at room temperature until completely dissolved. Glycerol (0.6%, *v/v*) was added to the chitosan solution as a plasticizer, after which FEO and PEO (1%, *v/v*) were added and sufficiently stirred to prepare 1% Chitosan-FEO, Chitosan-PEO, and Chitosan-F/P film-forming solution. Chitosan film-forming solution without FEO and PEO was used as the blank group. After the bubbles in film-forming solution were ultrasonically removed from the film-forming solution for 30 min, 20mL of the film-forming solution was poured into a plastic plate and dried at room temperature. The prepared Chitosan, Chitosan-FEO, Chitosan-PEO, and Chitosan-F/P composite films were stored in a desiccator (relative humidity of 53%) for 48 h before further experiments.

### 2.3. Film Thickness and Density

The flat, dry films were randomly selected from six different locations, and their thicknesses were measured using a vernier caliper (Mitutoyo, Kanagawa, Japan). The film volumes were calculated using the film areas and thicknesses, after which the density of the films was calculated via the mass and volume. The measurements were performed using the following equation:

$$\text{Density } (\text{g/cm}^3) = \frac{\text{weight } (\text{g})}{\text{thickness } (\text{cm}) \times \text{area } (\text{cm}^2)} \tag{1}$$

### 2.4. Water Swelling Assay

The water swelling ability of the films was determined using the method of Moradi et al. [12] The Chitosan, Chitosan-FEO, Chitosan-PEO, and Chitosan-F/P composite films ($2 \times 2$ cm$^2$) were weighed and immersed in a 100 mL beaker containing 50 mL distilled water, sealed with plastic wrap, and stored at 25 °C for 24 h. Then, the films were taken out and the surface water was quickly removed, after which they were weighed. The measurements were conducted using the following equation:

$$\text{Water swelling (\%)} = \frac{W_2 - W_1}{W_2} \times 100 \tag{2}$$

where $W_2$ was the weight of the swollen film (g) and $W_1$ was the weight of the dry film (g).

### 2.5. Water Solubility Assay

The procedure of the water solubility assay was consistent with the water swelling assay, except that the film was dried to constant weight at 105 °C after absorbing water. The water solubility of the film was determined using formula as bellow:

$$\text{Water solubility (\%)} = \frac{W_1 - W_2}{W_1} \times 100 \tag{3}$$

where $W_1$ was the initial weight of the film (g) and $W_2$ was the finial weight of the dry film (g).

### 2.6. Moisture Content Assay

The film was cut into $2 \times 2$ cm$^2$ and weighed, after which it was dried in an oven at 105 °C. The moisture content of the film was assessed by calculating weight loss after drying at 105 °C for 24 h. The moisture content of the films was calculated using the following formula:

$$\text{Moisture content} = \frac{M_1 - M_2}{M_1} \times 100 \tag{4}$$

where $M_1$ was the initial constant weight of the film and $M_2$ was the final constant weight of the film.

### 2.7. Film Color Assay

The color values of the films were evaluated using a Minolta colorimeter (CR-300, Minolta Camera Co., Osaka, Japan). The film was cut into 5 cm $\times$ 5 cm, six points at different locations on the film were measured, and each point was repeated 5 times. The L (blackness–whiteness), a (negative-green; positive-red), and b (negative-blue; positive-yellow) values of the film were determined. The total color difference ($\Delta E$) and whiteness values (WI) were shown in the following formulas:

$$\Delta E = \sqrt{(a^* - a)^2 + (b^* - b)^2 + (L^* - L)^2} \tag{5}$$

$$\text{WI} = 100 - \sqrt{(100 - L)^2 + a^2 + b^2} \tag{6}$$

The color parameter values of the standard plate were $L^*$ ($L^* = 98.09$), $a^*$ ($a^* = 0.40$), and $b^*$ ($b^* = 1.02$), and the color parameter values of the films were $L$, $a$, and $b$.

### 2.8. Fourier Transform Infrared Spectroscopy (FTIR) Assay

The interaction between the chitosan, FEO, and PEO, respectively, was observed by FTIR spectra. The films were cut into 2 cm $\times$ 2 cm and recorded using a Nicolet iS10 spectrometer (Thermo Fisher Scientific Inc., Waltham, MA, USA) equipped with attenuated total reflection (ATR) parts. The spectral resolution was 4 cm$^{-1}$ with 16 scans in a range of 400–4000 cm$^{-1}$.

*2.9. Thermal Stability Assay*

The differential scanning calorimetry (DSC) spectra of the films were measured using a Q2000 DSC system (TA Instruments, Newcastle, DE, USA). The films (10 mg) were placed onto a standard aluminum plate, and then heated at a rate of 20 °C/min. The heating temperature ranged from 0 to 350 °C.

*2.10. Environmental Scanning Electron Microscopy (ESEM)*

The surface and cross-section micrographs of the films were observed by an FEI-Quanta 200 ESEM (Hillsboro, Washington, DC, USA) at an accelerating voltage of 30 kV. The film was coated with gold and then observed. Both the surface and cross-section images were magnified 1000 times.

*2.11. Antioxidant Activity Assay*

The antioxidant activity of the Chitosan-FEO, Chitosan-PEO, Chitosan-F/P films, and Chitosan films were evaluated by measuring the scavenging capacity of the 2,2-diphenyl-1-picrylhydrazyl (DPPH) free radical. The DPPH radical scavenging assay was assessed using the method described by Blois et al. (1958) [13] with some modifications. The films were dissolved in ethanol and 1 mL of the complex solution was mixed with 1 mL of DPPH (dissolved in methyl alcohol, 0.1 mM), and incubated in the darkness at room temperature for 30 min. The UV absorbance of the samples was measured at 517 nm and the DPPH radical scavenging capacity of the samples was calculated using the following equation:

$$\text{DPPH scavenging activity } (\%) = [1 - \frac{\text{OD}_{570}(\text{sample})}{\text{OD}_{570}(\text{control})}] \times 100 \tag{7}$$

The $\text{OD}_{570}$ (control) was the absorbance value of the methanol solution of DPPH at 517 nm.

*2.12. Statistical Analysis*

All the assays were repeated in triplicate. The results were presented as mean ± standard deviation (SD). The statistical difference was calculated by the analysis of variance (ANOVA) and considered significant at $p < 0.05$ using the IBM SPSS ver. 22.

## 3. Results and Discussions

*3.1. Film Thickness and Density*

The effects of FEO and PEO on the thickness and density of the chitosan film are shown in Table 1. The incorporation of FEO and PEO affected the thickness and density of the chitosan film. The incorporation of FEO and PEO caused the thickness of the films to increase slightly, among which the Chitosan-FEO and Chitosan-F/P films were the most significant. Although there was no statistical difference in the test results, a clear trend was evident, indicating that the amount and type of EOs added correlated with the film thickness and density. The composite films were thicker than the pure chitosan films due to higher solid content per surface unit at a specific surface area [14]. Moreover, the cross-linking of polyphenols in FEO and PEO led to an increase in the thickness of the composite films [15]. Tan et al. [15] reported that a similar increasing trend occurred when grape seed extract was added to chitosan-based films.

**Table 1.** Thickness and density of Chitosan, Chitosan-FEO, Chitosan-PEO, and Chitosan-F/P films.

| Films | Film Thickness (mm) | Density (g/cm$^{-3}$) |
|---|---|---|
| Chitosan | 0.0696 ± 0.0016 | 0.0993 ± 0.0011 |
| Chitosan-FEO | 0.0679 ± 0.0009 | 0.1041 ± 0.0008 |
| Chitosan-PEO | 0.0686 ± 0.0011 | 0.102 ± 0.002 |
| Chitosan-F/P | 0.0699 ± 0.0011 | 0.1027 ± 0.0003 |

### 3.2. Film Moisture Content, Water Swelling, and Solubility

The water sensitivity of composite films was assessed via moisture content, water swelling, and water solubility. As shown in Table 2, the moisture content levels in the composite films varied according to the content and the nature of the FEO and PEO. With an increase in the EO concentration, the moisture content of the films decreased, and the Chitosan-FEO film displayed a lower water content level than the Chitosan-PEO film. The interaction between the chitosan and the phenols in FEO and PEO reduced the availability of hydroxyl and amino groups, while also reducing the hydrogen bond interactions between the chitosan and the water. Consequently, the moisture content of the chitosan composite films decreased [3]. The solubility of the composite films in an aqueous environment was a pivotal indicator for observing its water resistance and was also essential for evaluating the biodegradability of composite films as food packaging material [16]. The water swelling capacity and solubility of the composite films decreased in conjunction with an increase in the EO content, while the Chitosan-PEO film exhibited higher water swelling and solubility than the Chitosan-FEO film. The decrease in the water swelling and solubility of the composite films was possibly caused by the hydrophobicity of the FEO and PEO. Ojagh et al. [6] prepared the chitosan-based film by incorporating cinnamon EO, which was revealed to effectively reduce the moisture content and solubility of the composite films.

**Table 2.** Moisture content, water swelling, and solubility of Chitosan, Chitosan-FEO, Chitosan-PEO, and Chitosan-F/P films.

| Films | Moisture Content (%) | Water Swelling (%) | Water Solubility (%) |
|---|---|---|---|
| Chitosan | 17.2 ± 0.2 a | 660.8 ± 0.2 a | 27.80 ± 0.14 a |
| Chitosan-FEO | 16.34 ± 0.03 b | 365..83 ± 0.17 c | 21 ± 3 b |
| Chitosan-PEO | 17.2 ± 0.5 a | 431 ± 6 b | 22 ± 3 b |
| Chitosan-F/P | 15.46 ± 0.11 c | 312 ± 2 d | 21.0 ± 0.2 b |

Different letters in the same column indicate significantly different ($p < 0.05$).

### 3.3. Film Color Properties

The effect of the addition on the FEO and PEO on the color indicators of the film is shown in Table 3, significantly ($p > 0.05$) affecting the appearance of the composite films. The *L* and WI values of the Chitosan film were 71.78 ± 0.01 and 71.57 ± 0.01, while the *L* and WI values of composite films were higher than that of the Chitosan film, indicating that the FEO and PEO could significantly improve the brightness of the films. The *b* and Δ*E* values of the Chitosan-PEO film were the highest at 9.44 ± 0.01 and 77.02 ± 0.01, respectively, indicating that changes in the appearance of the Chitosan-PEO film were the most distinct. Moreover, the b value of all the composite films exceeded that of the Chitosan films and was an indicator of the tendency towards yellowness. The data were consistent with the physical appearance of the Chitosan, Chitosan-FEO, Chitosan-PEO, and Chitosan-F/P films. Changes in the color indicators could help avoid oxidative spoilage in packaged foods caused by visible and ultraviolet radiation, resulting in discoloration, off-flavors, and nutritional loss [17]. In our previous study [18], we found that the chitosan-based film containing syringic acid became dark and brownish-yellow.

**Table 3.** Color parameters including *L*, *a*, *b*, Δ*E*, and WI of the Chitosan, Chitosan-FEO, Chitosan-PEO, and Chitosan-F/P films.

| Films | *L* | *a* | *b* | Δ*E* | WI |
|---|---|---|---|---|---|
| Chitosan | 71.78 ± 0.01 [c] | −0.190 ± 0.001 [c] | 3.45 ± 0.01 [c] | 71.86 ± 0.01 [d] | 71.57 ± 0.01 [d] |
| Chitosan-FEO | 76.26 ± 0.13 [b] | −0.16 ± 0.01 [b] | 4.62 ± 0.02 [b] | 76.40 ± 0.13 [c] | 75.81 ± 0.13 [b] |
| Chitosan-PEO | 76.43 ± 0.01 [a] | −1.08 ± 0.01 [d] | 9.44 ± 0.01 [a] | 77.02 ± 0.01 [a] | 74.59 ± 0.01 [c] |
| Chitosan-F/P | 76.46 ± 0.03 [a] | 0.37 ± 0.01 [a] | 4.61 ± 0.01 [b] | 76.60 ± 0.03 [b] | 76.01 ± 0.03 [a] |

Different letters in the same column indicate significantly different ($p < 0.05$).

### 3.4. FTIR Analysis

The molecular interactions between chitosan and FEO and PEO, respectively, were reflected by FTIR spectra. As presented in Figure 1, the peaks at 3000 to 3500 cm$^{-1}$ were associated with the overlaps between the stretching vibration of the O–H group and the asymmetric and symmetric stretching of the N–H bond of the amino group in the same region [1,19]. In the control group, the peak strength of the Chitosan film at 3000 nm to 3500 cm$^{-1}$ was higher than that of the composite films incorporated with FEO and PEO, which could be attributed to the interaction of functional groups in FEO and PEO ingredients with –OH or –NH$^2$ of chitosan molecules, resulting in reduced O–H and N–H stretching [20]. The peaks at 1400 and 1530 cm$^{-1}$ are related to the in-plane bending of the O–H bond [1]. With the addition of FEO and PEO, these two peaks become flatter. Specifically, the Chitosan-P/F film peaks at 1400 and 1530 nm became less discernible due to the concentration and type of EOs. The peaks at 1640, 1260, and 650 cm$^{-1}$ were related to the C=C stretching, C–O stretching, and H-atom in the aromatic ring [21,22], representing the presence of phenolic compounds in the FEO and PEO in the composite films. The main compound in FEO was (E)-anethole, and the major biological substances in PEO were menthol and menthone. Furthermore, the FTIR spectra of each film were similar, indicating that no chemical reaction occurred that produced new substances. The results of FTIR spectroscopy explained the intermolecular interaction and molecular compatibility between the functional groups in FEO and PEO, as well as hydroxyl and amino groups in the chitosan chain [23].

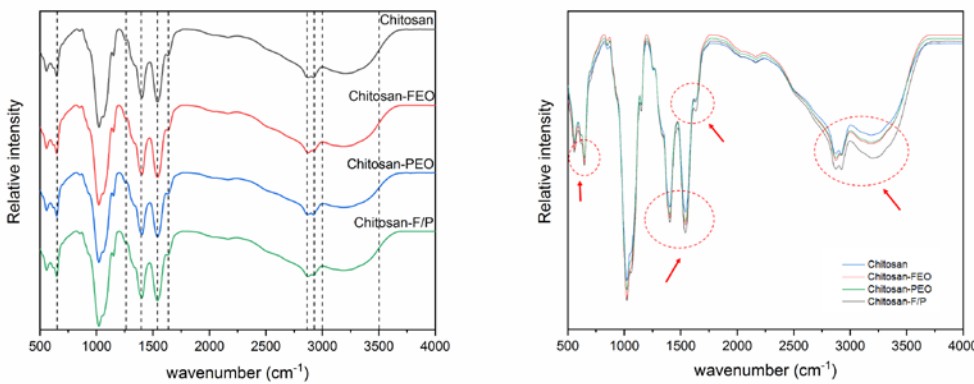

**Figure 1.** FTIR spectra of Chitosan, Chitosan-FEO, Chitosan-PEO, and Chitosan-F/P films.

### 3.5. DSC Analysis

The peak melting temperature of the films was measured via DSC analysis. DSC thermograms of the Chitosan, Chitosan-FEO, Chitosan-PEO, and Chitosan-F/P films are shown in Figure 2. The broad endothermic peak at approximately 150 °C was associated with the moisture composition, while the exothermic peak at 300 °C was related to the depolymerization and pyrolytic decomposition of the polysaccharide backbone [21]. The evaporation of the residual solvent caused the endothermic effect represented by the peak at 150 °C during the film preparation process [23]. Due to its high thermal stability, the highly crystalline material required considerable energy to destroy the crystal structure [19]. When the FEO and PEO were added to the chitosan films, the temperature of the endothermic peak increased significantly, indicating that the crystallinity of the composite films increased. The peak area at 300 °C increased with the increase in EO concentration, indicating that the addition of EOs enhanced the thermal stability of the composite films, which could be ascribed to the chemical etching caused by the chemical bond breakage, chain scission, or chemical degradation of the macromolecules [24]. In general, the increased thermal stability of the composite films represented a higher crystalline structure caused by the intermolecular interactions of EOs and chitosan, which might also affect the mechanical properties of the composite films.

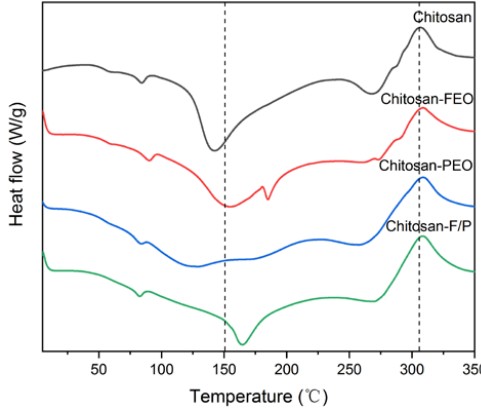

**Figure 2.** DSC spectra of Chitosan, Chitosan-FEO, Chitosan-PEO, and Chitosan-F/P films.

### 3.6. Microstructure Analysis

The ESEM images in Figure 3 show the surface and cross-sections of the Chitosan, Chitosan-FEO, Chitosan-PEO, and Chitosan-F/P films. The surface of both the Chitosan and composite films incorporated with FEO and PEO was smooth and homogenous without cracks, as in the previous study [18]. The uniform surface appearance of the films was caused by the presence of ordered and crystalline regions [25], indicating that the EOs and plasticizer (glycerol) could coexist compatibly at the molecular level. The cross-section of the Chitosan film appeared more regular, continuous, and compact compared with the composite film. When FEO and PEO were incorporated into the chitosan matrix, the composite films displayed an area of disconnection with small pores. The cross-sections of the Chitosan-FEO and Chitosan-PEO exhibited no significant differences. However, the cross-sections of the composite films containing FEO and PEO were the roughest of all the composite films, in which the EO droplets were distributed in the continuous polysaccharide network [26]. It is inferred that the EO concentration exerts a more substantial influence on the microstructure of the composite films than the type of EO. Sugumar et al. [27] prepared a eucalyptus oil nanoemulsion-impregnated chitosan film (NE–CH) at different concentrations, revealing that an increasing number of oil drops were tightly incorporated into the polymer matrix in conjunction with an increase in the nanoemulsion concentration of the eucalyptus oil.

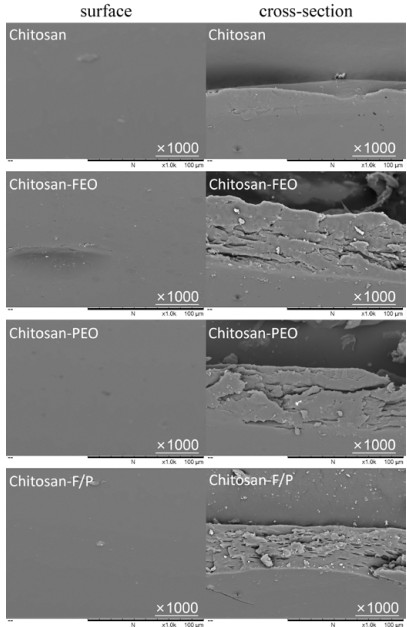

**Figure 3.** ESEM images of Chitosan, Chitosan-FEO, Chitosan-PEO, and Chitosan-F/P films.

### 3.7. Antioxidant Analysis

DPPH free radicals would be quenched and decolorized in the presence of antioxidants, resulting in a decrease in absorbance values [28]. The DPPH scavenging assay was used to determine the antioxidant activity of the films and to evaluate their antioxidant ability. The DPPH scavenging activity of Chitosan-FEO, Chitosan-PEO, and Chitosan-F/P films was assessed and presented in Figure 4. The antioxidant activity of the Chitosan film was the lowest (54.88%) of all the films. The free amino groups ($NH_2$) of chitosan reacted with hydrogen ions in a solution to form ammonium groups ($NH^{3+}$). Therefore, the antioxidant capacity of the Chitosan film was attributed to $NH_2$ groups of chitosan reacting with DPPH to form stable macromolecules [29]. The DPPH scavenging ability of the chitosan composite films with incorporated EOs was visibly improved ($p > 0.05$). The antioxidant activity of the Chitosan-PEO film (66.79%) was the lowest in the composite film but was significantly higher than that of the Chitosan film. Peppermint reportedly contains low to moderate levels of phenolics, which exhibit antioxidant activity [30]. The antioxidant activity of the Chitosan-F/P film was slightly higher than that of the Chitosan-FEO film, which was 68.21% and 69.09%, respectively, and could be ascribed to the dual antioxidant properties of FEO and PEO. The *trans*-anethole compound in FEO had a very prominent antioxidant effect [8].

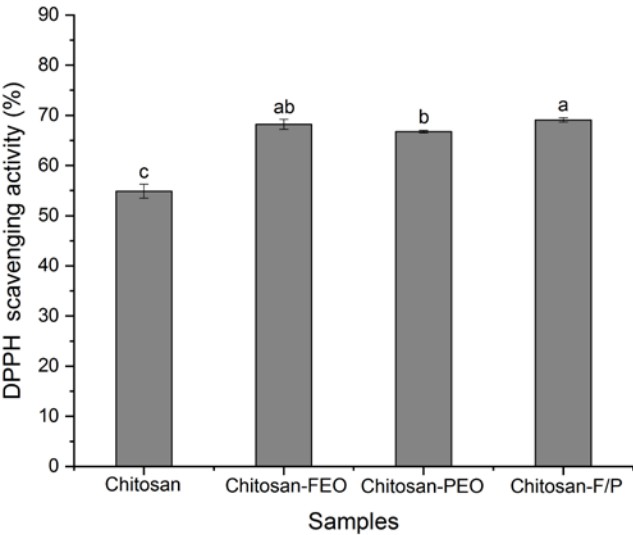

**Figure 4.** DPPH scavenging activities of Chitosan, Chitosan-FEO, Chitosan-PEO, and Chitosan-F/P films. Different letters in the same column indicate significantly different ($p < 0.05$).

### 4. Conclusions

This study indicates that active composite films with excellent properties can be prepared by adding FEO and PEO to chitosan-based films. The incorporation of the FEO and PEO can significantly enhance the barrier, physicochemical, and antioxidant properties of the composite films. The moisture content, water swelling, and solubility of the composite films decline, while the opacity and thermal stability increase. In addition, the color value changes (*L*, *a*, *b*, and Δ*E*) help to protect food from UV light-induced degradation. FTIR analysis shows that the improvement of the various properties of the composite films is caused by the interaction between the functional groups of the phenolic substances in the EOs and the chitosan. The chitosan-based composite films containing FEO and PEO, as a novel active packaging material, display considerable potential for the improvement of food packaging safety and shelf-life extension.

**Author Contributions:** Formal analysis, J.W.; investigation, F.C.; methodology, L.L.; resources, Z.T.; writing—review and editing, T.L. All authors have read and agreed to the published version of the manuscript.

**Funding:** This research was funded by Natural Science Foundation of Tibet Autonomous Region of China (Grant No. XZ2018ZRG-33(Z)).

**Conflicts of Interest:** The authors declare no conflict of interest.

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
