# Peer review of "Development and Characterization of Novel Active Chitosan Films Containing Fennel and Peppermint Essential Oils"

_coatings, doi:10.3390/coatings10100936_

Round 1
Reviewer 1 Report
The paper approach an interesting subject. However there is no novelty since the chitosan films with essential oil is largely described in literature. There are some issues which have to be explained in order to be published:
Why the fennel and peppermint? Why this combination?
Why this particular concentration of essential oil and chitosan? In the case of dual essential oil film which is the final concentration : 0.5 % or is 1 % for each EO
Composition of EO from producer or from authors analysis must be provided.
Why mechanical properties are not presented on the paper
Author Response
List of responses
Dear Editors and Reviewers
Thank you very much for your valuable comments concerning our manuscript entitled “Development and characterization of novel active chitosan films containing fennel and peppermint essential oils” (Manuscript ID: coatings-920754). These comments are all valuable and very helpful in revising and improving our manuscript, and have the important guiding significance to our study. We have studied comments carefully and made correction which we hope to meet with approval. Revised portions are marked in red in the manuscript. Attached please find the revised version, which we would like to submit for your kind consideration. The main corrections in the manuscript and the responds to the reviewer’s comments are as flowing:
Reviewer # 1:
- Why the fennel and peppermint? Why this combination?
Response: At present, there are few studies on chitosan films incorporated fennel essential oil (FEO) and peppermint essential oil (PEO). FEO is commonly used as the flavoring agents, cosmetics and pharmaceutical products. Some published research reported that biodegradable chitosan films incorporated with FEO displayed an excellent effect on antibacteria, antioxidant and preservation. The application of PEO is growing in popularity due to its exceptional sensory properties. PEO contains menthol, menthone and other major bioactive substances, exhibiting various inhibitory effects on foodborne pathogenic bacteria and fungi with different principal components, which can effectively prolong the shelf life of food products. What’s more, the purpose of this study is to explore the synergistic effect of FEO and PEO on the various properties of chitosan composite films.
- Why this particular concentration of essential oil and chitosan? In the case of dual essential oil film which is the final concentration : 0.5 % or is 1 % for each EO.
Response: The concentrations of EOs and chitosan used in the study were determined by multiple experiments and references, which is guaranteed to be effective in the various properties of chitosan composite films.
- Composition of EO from producer or from authors analysis must be provided.
Response: Special thanks to you for your comment. We added the composition analysis of EOs to the supplementary material.
- Why mechanical properties are not presented on the paper.
Response: Thank you very much for your good comment. The purpose of this study is to compare the differences on physicochemical property and determine the antioxidant effects of chitosan films incorporating FEO and TEO. In fact, a large number of research have reported that the chitosan-based films containing FEO and TEO have the prominent mechanical property. Thus, considering the structure of the manuscript, we do not evaluate the mechanical property of the composite films.
We appreciate for editor and reviewers’ warm work earnestly, and hope that the correction will meet with approval. Once again, thank you very much for your comments and suggestions.
Thank you and best regards.
Yours sincerely,
Liu Liu
Prof. of Shaanxi Normal University

Reviewer 2 Report
This manuscript is focused on preparation and the characterization of chitosan films containing fennel and peppermint essential oils. Generally, the paper can be published in Coatings; however, the presentation and discussion of the experimental results must be improved addressing the following comments:
Section 2.1. Materials: The chitosan must be thoroughly characterized regarding its molecular weight (by viscometry, light scattering, or size exclusion chromatography) and the degree of deacetylation (by NMR, IR, elemental analysis, or titration). The properties of chitosan are very dependent on these parameters; therefore the European Chitin Society does not recommend publication of the research papers on chitosan, in which this information is missing.
I would suggest the authors to provide additional data regarding mechanical properties of the films as these properties are essential for such materals.
Figure 1: The FTIR spectra for all four samples are absolutely identical. Please double-check.
Figure 3: Add the scale bars for all SEM images.
Table 2: The standard deviation should be expressed as ONE significant figure; that is, unless the number is between 11 and 19 times some power of ten, in which case you can use two significant figures. The mean value should be rounded off at the decimal place corresponding to the last significant digit of its standard deviation. E.g., 430.70 ± 5.56 should be presented as 431 ± 6, 22.24 ± 2.91 should be presented as 22 ± 3, etc.
Author Response
List of responses
Dear Editors and Reviewers
Thank you very much for your valuable comments concerning our manuscript entitled “Development and characterization of novel active chitosan films containing fennel and peppermint essential oils” (Manuscript ID: coatings-920754). These comments are all valuable and very helpful in revising and improving our manuscript, and have the important guiding significance to our study. We have studied comments carefully and made correction which we hope to meet with approval. Revised portions are marked in red in the manuscript. Attached please find the revised version, which we would like to submit for your kind consideration. The main corrections in the manuscript and the responds to the reviewer’s comments are as flowing:
Reviewer # 2:
- Section 2.1. Materials: The chitosan must be thoroughly characterized regarding its molecular weight (by viscometry, light scattering, or size exclusion chromatography) and the degree of deacetylation (by NMR, IR, elemental analysis, or titration). The properties of chitosan are very dependent on these parameters; therefore the European Chitin Society does not recommend publication of the research papers on chitosan, in which this information is missing.
Response: Special thanks to you for your good comments. We are very sorry for our negligence. The information regarding chitosan has been supplemented in the manuscript.
- I would suggest the authors to provide additional data regarding mechanical properties of the films as these properties are essential for such materals.
Response: Thank you very much for your good comment. The purpose of this study is to compare the differences on physicochemical property and determine the antioxidant effects of chitosan films incorporating FEO and TEO. In fact, a large number of researches have reported that the chitosan-based films containing FEO and TEO have the prominent mechanical property. Thus, considering the structure of the manuscript, we do not evaluate the mechanical property of the composite films.
- Figure 1: The FTIR spectra for all four samples are absolutely identical. Please double-check.
Response: In fact, the FTIR spectra are too similar caused by the ordinate. Thus, we have offered another spectra for your reference.
- Figure 3: Add the scale bars for all SEM images.
Response: Special thanks to you for your good comments. The scale bars have been added in the SEM images.
- Table 2: The standard deviation should be expressed as ONE significant figure; that is, unless the number is between 11 and 19 times some power of ten, in which case you can use two significant figures. The mean value should be rounded off at the decimal place corresponding to the last significant digit of its standard deviation. E.g., 430.70 ± 56 should be presented as 431 ± 6, 22.24 ± 2.91 should be presented as 22 ± 3, etc.
Response: Special thanks to you for your good comments. It has been changed in the Table 2.
We appreciate for editor and reviewers’ warm work earnestly, and hope that the correction will meet with approval. Once again, thank you very much for your comments and suggestions.
Thank you and best regards.
Yours sincerely,
Liu Liu
Prof. of Shaanxi Normal University

Reviewer 3 Report
Manuscript seems to be original and presents the physical, characterization and antioxidant properties of Chitosan-FEO, Chitosan-PEO and chitosan composite
films added with both two EOs (Chitosan-F/P) to identify the potential to be used as active packaging for fresh-cut meat or vegetable. The manuscript has certain novelty, even though chitosan is a known compound for encapsulation and as an active coating film, in a theme that is important for food consumers and not only, and is rather well written and presented. However, there are some queries that authors need to reply in order for their paper to be reconsidered for publication, since at this stage it cannot be accepted, please reply to the following:
- Concerning preparation of films, section 2.2. is this an in house method? if not please add reference. During sonication for bubble elimination, did authors controled temperature elevation within 30 min? Can increased temperature affect films?
- section 2.4.: authors say that the surface water was quickly from films (l. 87-88), how was it done?
- section 2.8: please describe in brief how the measurement was performed, was it an in house method? if not add reference/s, if yes include validation.
- concerning DPPH method, why samples dissolved in ethanol and not in methanol as the DPPH reagent?
- at conclusion authors say that: In addition, the change of color values (L, a, b, ΔE) was helpful to protect food from UV light-induced degradation. How can you be so sure since there were no experiments in the current work with a specific food?
- Section 3.4 FTIR, authors say that they believe in presence of the phenolic compounds in the FEO and PEO in the composite films. Essential oils do not contain phenolic compounds, usually extracts do. Please comment on this and explain which phenolics can be present, the same do at conclusion section where phenolics are also mentioned. Probably authors mean phenols like thymol, carvacrol, eugenol, if this is the case please rephrase.
- which film authors believe is the best after this study that could be used in the food industry? It would be interesting if authors have checked the studied films with a food matrix and have evidence about the increase in shelf-life and safety as mentioned at conclusion, how are you sure about increase in shelf life?
- minor spelling and grammar mistakes.
Author Response
List of responses
Dear Editors and Reviewers
Thank you very much for your valuable comments concerning our manuscript entitled “Development and characterization of novel active chitosan films containing fennel and peppermint essential oils” (Manuscript ID: coatings-920754). These comments are all valuable and very helpful in revising and improving our manuscript, and have the important guiding significance to our study. We have studied comments carefully and made correction which we hope to meet with approval. Revised portions are marked in red in the manuscript. Attached please find the revised version, which we would like to submit for your kind consideration. The main corrections in the manuscript and the responds to the reviewer’s comments are as flowing:
Reviewer # 1:
- Concerning preparation of films, section 2.2. is this an in house method? if not please add reference. During sonication for bubble elimination, did authors controled temperature elevation within 30 min? Can increased temperature affect films?
Response: The methods regarding prepared films refer to our previous published paper with slight modification (Yang et al., 2019). We treat the film-forming solution with ultrasound at room temperature to ensure that the temperature would not affect the film-forming solution.
Ke Yang, Hui Dang, Liu Liu, Xinzhong Hu, Xiaoping Li, Zhen Ma, Xiaolong Wang, Tian Ren. Effect of syringic acid incorporation on the physical, mechanical, structural and antibacterial properties of chitosan film for quail eggs preservation. International Journal of Biological Macromolecules. 141 (2019) 876-884.
- section 2.4.: authors say that the surface water was quickly from films (l. 87-88), how was it done?
Response: The film is sealed for 24 hours and the surface moisture of films is gently wiped with degreasing cotton for further determination.
- section 2.8: please describe in brief how the measurement was performed, was it an in house method? if not add reference/s, if yes include validation.
Response: The FTIR is determined according to the method of Yang et al. The prepared films are cut into small strips and measured by a Nicolet iS10 spectrometer equipped with attenuated total reflection (ATR) parts.
- concerning DPPH method, why samples dissolved in ethanol and not in methanol as the DPPH reagent?
Response: Many published literatures regard ethanol as the best solvent to dissolve films. Ethanol is used as an organic solvent to dissolve the films and determine their antioxidant property. Your suggestion provides us with a novel idea, and we can carry out supplementary verification in the future research.
- at conclusion authors say that: In addition, the change of color values (L, a, b, ΔE) was helpful to protect food from UV light-induced degradation. How can you be so sure since there were no experiments in the current work with a specific food?
Response: The previous research (Rubilar et al., 2013) indicated that the decreases of film lightness may help to avoid oxidative deterioration in packaged foods caused by exposure to visible and ultraviolet light, leading to nutrient losses, discoloration and off-flavours. Yang et al found that The fresh quail eggs coated with film-forming solution has a certain preservation effect.
Rubilar et al. Physico-mechanical properties of chitosan films with carvacrol and grape seed extract. Journal of Food Engineering. 115 (2013) 466-474.
- Section 3.4 FTIR, authors say that they believe in presence of the phenolic compounds in the FEO and PEO in the composite films. Essential oils do not contain phenolic compounds, usually extracts do. Please comment on this and explain which phenolics can be present, the same do at conclusion section where phenolics are also mentioned. Probably authors mean phenols like thymol, carvacrol, eugenol, if this is the case please rephrase.
Response: Thank you very much for your suggestion. We have made modifications in the manuscript.
- which film authors believe is the best after this study that could be used in the food industry? It would be interesting if authors have checked the studied films with a food matrix and have evidence about the increase in shelf-life and safety as mentioned at conclusion, how are you sure about increase in shelf life?
Response: In this study, the Chitosan-F/P composite film has the best properties among all the composite films. Our study testified that the composite films incorporated FEO and PEO have prominent antioxidant activity, and the physical and chemical properties were also significantly improved. What’s more, FEO and PEO have a certain antibacterial effects. Thus, we infer that the chitosan films incorporated FEO and PEO can effectively extend the shelf life of food. In future research, we will further verify the preservation effect of Chitosan-FEO, Chitosan-PEO, and Chitosan-F/P composite film on food.
- minor spelling and grammar mistakes.
Response: The manuscript is carefully edited by a native English speaker and we attach a certificate.
We appreciate for editor and reviewers’ warm work earnestly, and hope that the correction will meet with approval. Once again, thank you very much for your comments and suggestions.
Thank you and best regards.
Yours sincerely,
Liu Liu
Prof. of Shaanxi Normal University

Round 2
Reviewer 1 Report
The revised manuscript is improved and the paper can be published in your journal
Author Response
List of responses
Dear Editors and Reviewers
Thank you very much for your reply concerning our manuscript entitled “Development and characterization of novel active chitosan films containing fennel and peppermint essential oils” (Manuscript ID: coatings-920754). We appreciate for editor and reviewers’ warm work earnestly. Once again, thank you very much for your comments.
Thank you and best regards.
Yours sincerely,
Liu Liu
Prof. of Shaanxi Normal University
Reviewer 2 Report
Unfortunately, the authors did not take the revision of the paper seriously and did not take into account my comments and comments from other reviewers. I fear I cannot recommend this paper for publication.
The chitosan sample is insufficiently characterized. The properties of chitosan are very dependent on its molecular weight the degree of deacetylation; therefore, the wide ranges of values provided by the manufacturer (like the degree of deacetylation ≥ 90%, or molecular weight of around 1.5×105 Da) are clearly insufficient.
The mechanical properties of the films are essential for such materials. Without these properties, your study looks incomplete.
Section 3.4 and Figure 1: I want to emphasize once again that The FTIR spectra for all four samples are absolutely identical. Therefore, it is not clear to me what changes in the spectra are discussed by the authors. I cannot accept the conclusions drawn from these spectra (“The results of FTIR spectroscopy explained the intermolecular interaction and molecular compatibility between the functional groups in FEO and PEO, as well as hydroxyl and amino groups in the chitosan chain”). Section 3.4. must be completely revised.
Table 2: It is a pity that the authors do not know what the significant figures are. Please read about significant figures in any textbook on statistics or even in Wikipedia http://en.wikipedia.org/wiki/Significant_figures
Author Response
List of responses
Dear reviewer
Thank you for your letter and the comments concerning our manuscript entitled “Development and characterization of novel active chitosan films containing fennel and peppermint essential oils” (Manuscript Number: coatings-920754). We have substantially revised our manuscript after reading the comments and the revised portions are marked in red in the manuscript. The main corrections in this manuscript and the responds to the comments are as flowing:
- The chitosan sample is insufficiently characterized. The properties of chitosan are very dependent on its molecular weight the degree of deacetylation; therefore, the wide ranges of values provided by the manufacturer (like the degree of deacetylation ≥ 90%, or molecular weight of around 1.5×105 Da) are clearly insufficient.
Response: We are very sorry that our reply did not satisfy you. The chitosan used in this study is the same as the one in the previous published paper (Yang et al., 2019). We described the chitosan information according to the label.
- The mechanical properties of the films are essential for such materials. Without these properties, your study looks incomplete.
Response: Thank you very much for your comment. On the one hand, numerous papers have fully characterized the mechanical properties of chitosan films containing FEO and PEO. On the other hand, the chitosan films containing other EOs and chitosan-FEO and chitosan-PEO films have characterized their mechanical properties together and discussed the mechanism of their mechanical properties, this part of data will be presented in another manuscript. Thanks again for your comment, we will design a more complete scheme in the future research.
- Section 3.4 and Figure 1: I want to emphasize once again that The FTIR spectra for all four samples are absolutely identical. Therefore, it is not clear to me what changes in the spectra are discussed by the authors. I cannot accept the conclusions drawn from these spectra (“The results of FTIR spectroscopy explained the intermolecular interaction and molecular compatibility between the functional groups in FEO and PEO, as well as hydroxyl and amino groups in the chitosan chain”). Section 3.4. must be completely revised.
Response: We have offered another spectra for your reference, and a table regarding the relative intensity of characteristic peaks to illustrate the differences among different samples.
- Table 2: It is a pity that the authors do not know what the significant figures are. Please read about significant figures in any textbook on statistics or even in Wikipedia http://en.wikipedia.org/wiki/Significant_figures
Response: Special thanks to you for your good comments. It has been changed in the Tables.
We appreciate for editor and reviewers’ warm work earnestly, and hope that the correction will meet with approval. Once again, thank you very much for your comments and suggestions.
Thank you and best regards.
Yours sincerely,
Liu Liu
Prof. of Shaanxi Normal University

Reviewer 3 Report
Authors replied in full to my comments and suggestios and i am satisfied with their response. Their manuscript has been significantly improved and can be now be published in its current form.
Author Response
List of responses
Dear reviewer
Thank you very much for your reply concerning our manuscript entitled “Development and characterization of novel active chitosan films containing fennel and peppermint essential oils” (Manuscript ID: coatings-920754). We appreciate for editor and reviewers’ warm work earnestly. Once again, thank you very much for your comments.
Thank you and best regards.
Yours sincerely,
Liu Liu
Prof. of Shaanxi Normal University
Round 3
Reviewer 2 Report
- Please add the attached FTIR spectra and Table with FTIR data to the main body of the manuscript. Otherwise, the discussion and conclusions are not clear for readers.
- I am sincerely surprised that the authors cannot figure out the significant numbers. Once again: The standard deviation should be expressed as ONE significant figure; that is, unless the number is between 11 and 19 times some power of ten, in which case you can use two significant figures. The mean value should be rounded off at the decimal place corresponding to the last significant digit of its standard deviation.
You can find the corrected data in the attachment. Please make appropriate corrections in the Tables and the data in the text.

Author Response
List of responses
Dear reviewer
Thank you for your letter and the comments of reviewer and editor concerning our manuscript entitled “Development and characterization of novel active chitosan films containing fennel and peppermint essential oils” (Manuscript Number: coatings-920754). We have substantially revised our manuscript after reading the comments and the revised portions are marked in red in the manuscript. The main corrections in this manuscript and the responds to the comments are as flowing:
Reviewer # 2:
- Please add the attached FTIR spectra and Table with FTIR data to the main body of the manuscript. Otherwise, the discussion and conclusions are not clear for readers.
Response: Thanks for your comment. We have added the attached FTIR spectra to the original manuscript.
- I am sincerely surprised that the authors cannot figure out the significant numbers. Once again: The standard deviation should be expressed as ONE significant figure; that is, unless the number is between 11 and 19 times some power of ten, in which case you can use two significant figures. The mean value should be rounded off at the decimal place corresponding to the last significant digit of its standard deviation.
Response: Thank you very much for your comment. We have corrected the significant numbers according to your instructions.
We appreciate for your warm work earnestly, and hope that the correction will meet with approval. Once again, thank you very much for your comments and suggestions.
Thank you and best regards.
Yours sincerely,
Liu Liu
Prof. of Shaanxi Normal University